# Enhancing integrated analysis of national and global goal pursuit by endogenizing economic productivity

**Barry B. Hughes**[1][☉]*, **Kanishka Narayan**[2][☉]

1 Frederick S. Pardee Center for International Futures, Joseph Korbel School of International Studies, University of Denver, Denver, Colorado, United States of America, 2 Pacific Northwest National Laboratory, Joint Global Change Research Institute, College Park, Maryland, United States of America

☉ These authors contributed equally to this work.
* bhughes@du.edu

**Data Availability Statement:** All relevant data are within the manuscript and its Supporting Information files.

**Funding:** The authors received no specific funding for this work.

## Abstract

Analysis with integrated assessment models (IAMs) and multisector dynamics models (MSDs) of global and national challenges and opportunities, including pursuit of Sustainable Development Goals (SDGs), requires projections of economic growth. In turn, the pursuit of multiple interacting goals affects economic productivity and growth, generating complex feedback loops among actions and objectives. Yet, most analysis uses either exogenous projections of productivity and growth or specifications endogenously enriched with a very small set of drivers. Extending endogenous treatment of productivity to represent two-way interactions with a significant set of goal-related variables can considerably enhance analysis. Among such variables incorporated in this project are aspects of human development (e.g., education, health, poverty reduction), socio-political change (e.g., governance capacity and quality), and infrastructure (e.g. water and sanitation and modern energy access), all in conditional interaction with underlying technological advance and economic convergence among countries. Using extensive datasets across countries and time, this project broadly endogenizes total factor productivity (TFP) within a large-scale, multi-issue IAM, the International Futures (IFs) model system. We demonstrate the utility of the resultant open system via comparison of new TFP projections with those produced for Shared Socioeconomic Pathways (SSP) scenarios, via integrated analysis of economic growth potential, and via multi-scenario analysis of progress toward the SDGs. We find that the integrated system can reproduce existing SSP projections, help anticipate differential economic progress across countries, and facilitate extended, integrated analysis of trade-offs and synergies in pursuit of the SDGs.

## Introduction

Alternative projections of economic growth—for individual counties, global regions, and the world—support multiple policy analysis objectives, including exploring poverty reduction or

**Competing interests:** The authors have declared that no competing interests exist.

climate change. For many specialized purposes, alternative exogenous assumptions about future growth can suffice.

Yet, significant causal dynamics through economic productivity and growth link interventions to achieve any Sustainable Development Goal (SDG) to progress toward multiple other goals. Such dynamics lie at the core of trade-offs and synergies related to pursuit of goals [1–6] —as do financial and physical resource constraints. This project differentiates and represents two-way relationships between productivity and driving forces as diverse as educational attainment (plus its quality) and health of populations, quality of governance (including levels of corruption and efficiency), extent of infrastructure, research and development expenditures, international trade and financial system openness, and climate change.

Despite significant foundational work on endogenizing productivity [7–10], model-based, long-term economic growth projection efforts with endogenization remain limited and tend to draw on select sets of drivers. Lejour *et al.* [11] looked to research and development spending, as did Capros *et al.* [12]; Kypreos and Bahn [13] used a learning curve approach; van der Mensbrugghe [14] identified the importance of export orientation; Lofgren and Diaz-Bonilla [15] looked at trade openness plus government capital spending.

Important steps toward more extensive endogenization of productivity in long-term modeling have been made. For example, the five shared socioeconomic pathway (SSP) scenarios [16–18] are supported by studies producing century-long, quantitative projections across countries for key variables including population, education, and urbanization. Two studies have produced standardized GDP projections for the SSPs [19, 20]. Both used methodologies that represented economic growth as functions of capital, labor, and TFP. They built on a basic representation of conditional convergence of countries to technological frontiers in interaction with driving forces such as educational attainment [19] or trade openness, regulatory barriers to market access, and energy demand [20]. Such standardized projections of growth contribute not just to analysis of mitigation of and adaptation to climate change, but to the exploration of a very broad range of the SDGs [4, 21].

At the country level the Millennium Institute's iSDG model includes detail on 78 indicators across the SDGs. It represents total factor productivity in agriculture, industry, and services with functions having multiple drivers, normalized and assuming Hicks-neutral technological change, a representation with some similarities to that elaborated here [22]. Applications like that for Tanzania by Collste, Pedercini, and Cornell [23] illustrate country-specific utility.

This article describes a modeling approach that considerably further extends the endogenization of productivity, imbedding the resultant structure in the economic model of the International Futures (IFs) integrated assessment system [24]. IFs is an extensive system of hard-linked models that, with these productivity formulations, dynamically represents two-way causal connections between endogenously-represented economic productivity change and numerous endogenously-represented driving/driven variables (more than 100 of which are either indicator variables associated with the SDGs or close relatives of them).

In general terms, this TFP modeling approach combines attention to a core pattern of technological advance by system leadership and conditional convergence to leading levels (Abramowitz [25] and Baumol [26] provided foundations for the theory of convergence with a representation of the manner in which a wide range of country-specific, policy-relevant variables modify that core pattern). IFs is a dynamic, annually recursive system that includes a general equilibrium economic model with a Cobb-Douglas production function. IFs represents 186 countries, has a base year of 2015, and runs through the 2030 horizon of the SDGs and on through the century. In all country-years TFP in the production function is a cumulative level or stock of productivity (incremented annually), interacting with the labor and

capital stock terms. The change in TFP affects many model dynamics, and most explanation here focuses on representation of its change.

Following the methods section, the results section illustrates what such an integrated analytical system can suggest about future productivity and growth in both the Base Case and in alternative scenarios. It demonstrates long-term analysis capability via comparison with SSP quantification, short-term analysis via insight into differential country growth potential, and mid-range analysis contribution via integrated analysis of SDGs across multiple scenarios.

## Methods

Methods subsections address four sequential analytical steps in building the specifications to represent the dynamics of TFP change by country-year:

1. Identifying and understanding TFP historical data. The project draws on two sources, the Conference Board and the Penn World Tables.

2. Representing the core pattern of TFP convergence, including the speed of change at the leading edge and the conditionally changing patterns of catch-up by following countries, including the typical slowing of those nearing the leaders.

3. Understanding the contribution to the conditional convergence process of many potentially policy-relevant variables. A significant challenge comes from both the frequently high levels of correlation within an unusually large set of drivers or independent variables (IVs) and between them and the GDP per capita variable that shapes the conditional convergence core change in TFP. Attention to economic growth and productivity literatures and to model behavior must supplement statistical analysis.

4. Building the model structure into the dynamic, recursive IFs system to represent the combined contributions of the basic convergence process and the multiple IVs, with attention to the speed of TFP changes in response to changes in drivers.

### Identifying and exploring TFP data

The Conference Board [27] and the Penn World Tables [28] provide TFP data (annual change and cumulative stock levels) for many countries and years. The Conference Board provides country-year data on the labor, capital, and productivity contributions to economic growth, produced by them via growth accounting for GDP.

The TFP series presented by the Penn World Tables (PWT) Release 9.0 is a ratio of TFP for individual countries to that of the United States, for which the value is set to 1. As elaborated in S1 Appendix, the authors used growth accounting in a Cobb-Douglas production function to calculate country-specific TFP stock series as the Solow residual (routinely explained in economic development texts) using PWT data on GDP, labor stocks, capital stocks, and the relevant exponents in a Cobb-Douglas function [29]. That created series of annual changes and cumulative levels/stocks of TFP from 1960 through 2014 for 127 countries.

Fig 1 illustrates historical patterns of change in the TFP stock series for five countries since 1960. Especially for South Korea, the figure shows the possibility of convergence toward system leaders. Technological convergence is, however, often much slower than is convergence in GDP per capita and the contribution of growing capital stocks to GDP per capita growth. For instance, the ratio of US to Chinese TFP in Fig 1 narrowed only from 3.6 to 3.2 between 1960 and 2015, while the ratio of US to Chinese capital stock changed dramatically from 14.7 to 0.76, illustrating the importance also of modeling investment (and, of course, the changing size of the labor force).

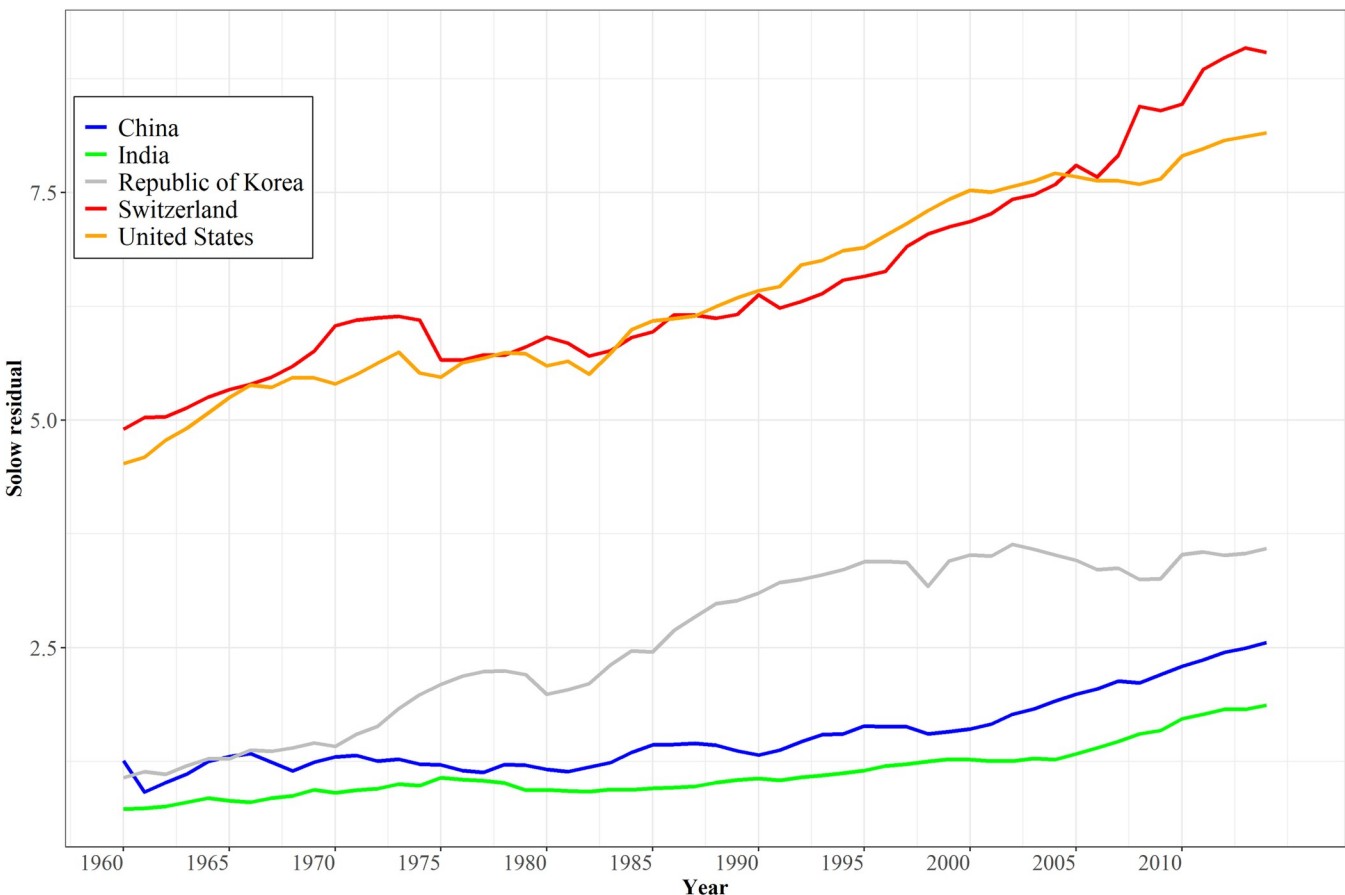

**Fig 1. The stock of TFP (Solow residual) in selected countries.** Note: The stock was computed by dividing GDP by the product of capital raised to the long-term average value of alpha (capital share of value added) and labor to 1 minus alpha. Source: IFs Version 7.61, using data from the Penn World Tables, release 9.0.

Fig 2 directs attention to the annual changes in the TFP series, looking across World Bank country-income groupings. Annual change fluctuates very substantially with a wide variety of short- and long-term influences including business cycles. Not only is there substantial short-term volatility in TFP even with smoothing via 10-year moving averages, but a long cycle is evident. TFP growth rates for high-income countries declined from at least 1970 until the early 1980s and in other income groupings until the early 1990s. Conference Board data also show this pattern.

The temporal pattern in TFP (Solow residual) change is very different from the steadier growth in most of the standard candidates for its drivers, such as annual years of adult educational attainment, educational quality, health of populations, extent of infrastructure, and even quality of governance. Further, volatility in driving variables such as life expectancy (for instance, in the face of factors such as the HIV/AIDS epidemic) or governance quality (subject to idiosyncrasies of leadership) seldom has more than tenuous relationship to such broad cyclical productivity change. The cycle appears much more likely to have been associated with general completion of post-World War II rebuilding and technological catch-up, the sharply rising prices of oil and natural gas and related financial flow and national debt challenges of the 1970s and 1980s, and the impacts of globalization acceleration and technological change in more recent decades. Durlauf, Johnson, and Temple ([30]: 624–625) make very similar points

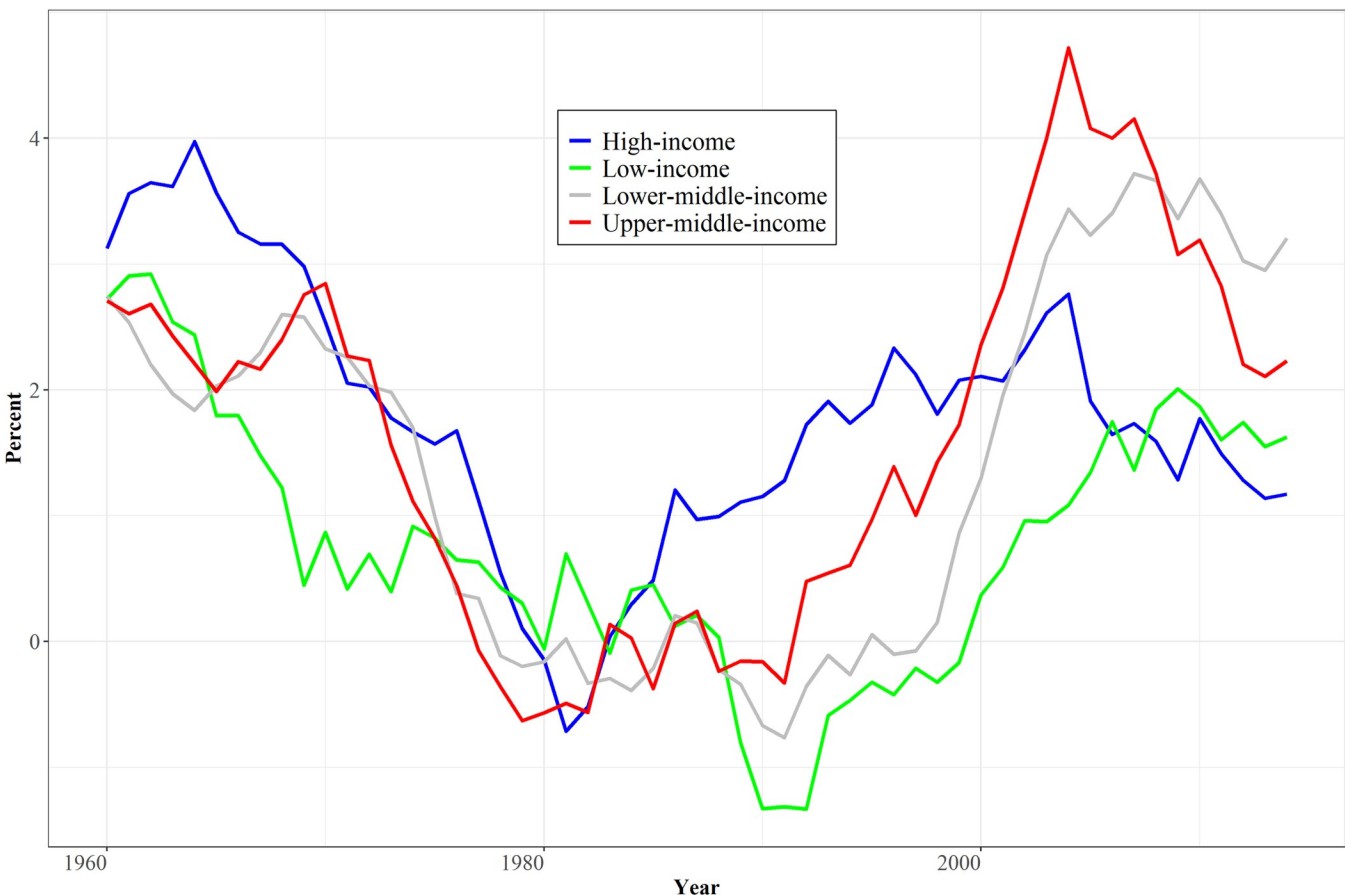

**Fig 2. Annual change of TFP in World Bank income categories.** Note: Solow residuals using 10-year moving averages; additional countries enter series over time, mostly at beginnings of decades. Source: IFs Version 7.61, building upon data from the Penn World Tables, release 9.0.

about the problems associated with time series analysis in understanding TFP change. Longitudinal analysis in this project found very poor relationships of IVs and TFP, often with the wrong sign; that pushed us toward cross-sectional analysis with careful attention to panels across time. The contributions from cross-sectional analysis assist in understanding TFP convergence across time.

## Representing technological leadership and basic convergence

Fig 1 showed the long-term pattern of TFP growth in system leaders such as the United States and Switzerland. Between 1960 and 2014 average annual growth in the U.S. was 1.1%. Countries not at the leading edge generate advance in levels of TFP less by innovating in either hard or soft technology, although both occur, than by adopting and adapting. That basic convergence process can be identified in the historical data patterns, including the catching up occurring within longitudinal data for countries such as South Korea. S3 Appendix contains cross-sectional analysis at 10-year intervals. The years since 2000 show the same convergence via downward-sloping lines in cross-sectional analysis of TFP change with GDP per capita. In 1960 and 1970, however, the relationship was upward sloping, indicating diverging productivity growth. The switch in slope reinforces the probable existence of an inverted-U pattern, with middle-income countries more capable of adopting and adapting or also innovating in hard and soft technology than lower-income ones; see Johnson and Papageorgiou [31] for

analysis of convergence including its temporal intermittency and variation across country clubs. Middle-income countries have come into their own quite strongly in more recent years. The switch may also reflect, however, the changing benefits and costs of globalization over a long time period, including the energy and debt crises of the 1970s and 1980s. The pattern of earlier years reinforced the conclusion of many writers in the political economy literature of the time, namely that, overall, rich and poor countries were diverging [32].

Core productivity formulations for long-term forecasting thus often begin with a specification of the rate of advance in a leading country and of the basic or conditional convergence process for other countries. That can also be usefully controlled exogenously for scenarios, reflecting the variation of its pattern over time. IFs uses the approach, with a default specification in Base Case scenario analysis of about 1% annual advance in the U.S. (some variation by sector) and a default inverted-U convergence rate as a function against GDP per capita at purchasing power parity (PPP) on top of that (with low-income countries at about 1% additional advance, lower-middle-income countries with annual increments of 2–3% and upper-middle income countries beginning to slow their rate of convergence as they catch higher-income ones). Overall, that corresponds roughly to what Barro [33] referred to as an iron law long-term rate globally of about 2% and is not very different from the values assumed in developing the SSP quantifications, especially SSP2, by Cuaresma [19] and Dellink, *et al.* [20]. The convergence increments across economic development levels were scaled for IFs by analysis of recent growth patterns and need be responsive to new data as those patterns change again, as well as to scenario assumptions.

Given that the pattern of convergence appears responsive to a very broad range of developmental factors, the core representation assumes implicitly, *ceteris paribus*, that those factors advance in roughly the same fashion over time and across counties. Convergence and, in fact, the rate of advance at the leading edge are, however, highly conditional upon the pattern of change in drivers that supplement GDP per capita. The atypical or unexpected levels of driving variables contribute an increment or decrement to advance of productivity in both reality and the model.

## Understanding multiple, interacting contributions to TFP change

To reiterate, this project's central purpose is to develop and use formulations that link a wide range of TFP drivers to annual change of TFP in a dynamic, recursive, integrated assessment model, facilitating analysis both of uncertainty and the impacts of a wide range of policy orientations and interventions on future productivity growth and on further to policy goals like the SDGs. Foundations already described have included measurement of TFP with explorations of its behavior historically and representation of a core pattern of conditional convergence in relationship with GDP per capita. Building on those foundations, it is important to explore the incremental relationship of a wide range of possible independent variables (IVs) with GDP per capita, with each other, and with TFP.

This central purpose has extensive methodological implications. Not least is the reality that representing an exceptionally wide range of drivers of TFP, often highly interactive with each other within and across countries and socio-economic development processes, poses significant challenges to statistical analysis. Within IFs currently there are 16 variables directly linked to change in TFP that are computed endogenously within the interaction of the many models in the system and additionally manipulable exogenously for uncertainty and policy analysis. See S4 Appendix for a survey of the IFs models and interactions. Analysis with IFs is focused on the implications of policy orientations or choices that affect single variables in that set, clusters of them, or the playing out of all 16 in scenarios attentive to still other important variables

in IFs (including in recent Pardee Center analysis of the impacts from COVID-19 for the UNDP; see https://data.undp.org/content/assessing-covid-impacts-on-the-sdgs/).

Further, changes in TFP feed forward to economic growth and on to the various subsystems that indirectly or directly drive those same variables affecting it. The temporal dynamics of the annually recursive system therefore become another important methodological consideration for the TFP formulation and parameterization. Those dynamics involve flow variables such as births, deaths, capital investment, and depreciation and stock variables such as population numbers by age and sex and capital stock—a small illustrative sample in just the population and economic models. TFP itself is a stock variable in IFs, changed annually by increments determined by those 16 drivers. Illustrating temporal complications in analysis, reduced stunting stemming from childhood undernutrition and increased years of adult educational attainment are highly correlated across countries; yet, changes in each resulting from different policy choices across countries play out over quite different time horizons, reinforcing our desire to represent both drivers of change in TFP.

Very large literatures explore the empirics of growth [34] and illustrate common approaches to analyzing it. Much analysis has been focused on the aggregate determination of GDP growth rather than differentiating labor, capital, and total factor productivity contributions to it (that differentiation being important because policy analysis benefits from analyzing the multiple drivers of production factors separately and in combination). When the focus has been productivity, much attention in endogenous growth theory and empirics has been on technology innovation and knowledge, often related to R&D and sometimes also human capital [35]; Comin and Mestieri [36] reviewed the long history of attention to factors influencing technology diffusion). Almost all literature has focused on single or small sets of determinants in longitudinal, cross-sectional, or panel analysis, facilitating common statistical approaches including the use of generalized method of moments with instrumental variables [37].

Statistical estimation challenges in analysis of even small sets of independent variables within dynamic systems include missing and poor data, measurement errors, multicollinearity, potential utility of instrumental variables, bi-directionality of causality, intercountry effects, differential and sometimes very long lag times, and alternative potential functional forms for relationships. Many standard approaches for addressing such issues do not work at all easily in the context of extensively integrated assessment modeling and its policy analysis, as undertaken in this project.

The statistical analysis here with the much larger set of IVs is fairly basic and very much subject to enhancement in the future. The methodology of this project necessarily combines attention to statistical analysis with attention to literature often more narrowly focused on specific drivers of productivity and growth and attention to temporal dynamics of interventions within the resulting IFs system. Qualitative assessment and judgment must supplement purely quantitative analysis.

**Basic structural and methodological considerations.** The structural development literature has long recognized that most developmental variables advance in rough relationship with each other and with GDP per capita (Chenery and Syrquin [38]; Chenery [39]; Kuznets [40]; Sachs [41]; and Syrquin and Chenery [42]). GDP per capita is almost invariably treated as the driving variable of the basic or conditional core TFP convergence formulation and as such is the general representative of structural socio-economic development.

Thus, the focus with respect to the conditional aspect of that convergence needs to be on the marginal impact on TFP of other drivers—the potential impact of those other drivers that is unrelated to the structural development pattern that GDP per capita represents. Thus, the IFs economic model looks to changes in the other IVs unexplained by GDP per capita to modify the conditional convergence calculation of TFP changes over time. Analysis relating GDP

per capita to other variables in the broader IV set represented in IFs found that logged GDP per capita provided the most explanatory power across functional forms. Hughes [43] elaborated the tendency for many potential IVs to saturate relative to continued advance of GDP per capita, providing further basis for logging GDP per capita in exploring its relationship with them.

This approach also addresses much of the multicollinearity within the full set of IVs. As already indicated, however, representation of so many IVs means that control for GDP per capita will not fully address it; as Table 2 will show, the residuals themselves are often correlated. Again, attention to more specialized literature and qualitative judgment must be part of the methodology. Future analysis will be able to enhance the statistical analysis by combining use of continually improving data with more sophisticated statistical approaches, including exploring the value that instrumental variables [44: Chapter 15] and non-linear control of the log of GDP per capita could add.

Another methodological issue is treatment of productivity drivers that are not inherently related to structural patterns of socio-economic development within specific countries. A critical one is climate change, which some IAMs such as DICE, PAGE and FUND link forward to annual levels of GDP but less frequently to the underlying production factors producing that GDP [45]. Stern [46] argued, however, that climate change could affect not just annual GDP, but both the capital accumulation and the advance of productivity stocks underlying its production; an OECD study [47: pp. 81–84], using a version of the DICE model, directed 30% of damage to TFP, significantly increasing end-of-century GDP reduction relative to typical analyses with the model. As with DICE's linkage of temperature to GDP, IFs uses a second-degree polynomial equation to link temperature to a multiplier on TFP; the interface allows flexible parameterization. A second driver largely independent from economic development and therefore not controlled by GDP per capita is trade integration with the global economy, linked to TFP within IFs via a linear formulation.

In scientific research and in explorations of specific policy lever impact, the emphasis is often parsimony. For policy analysis in domains like the extensive set of SDGs, however, and in any effort to understand national policy options over the longer run, there is need also for attention to many interacting variables. Hence, the attention to and representation of a wide set of IVs, despite the very significant methodological complications of doing so.

**The relationship of multiple drivers to TFP.** Statistically, our interest is in the relationship between TFP and the residuals of IVs in their own relationship with logged GDP per capita. Because growth in TFP is longitudinally unrelated to change in many developmental variables, and because cross-sectional patterns have changed over time, related in part to the movement from divergence to convergence in global TFP patterns, Table 1 shows the cross-sectional relationships for selected IVs at different time points. S2 Appendix documents the IVs examined. All independent and dependent variables series are open for use by others via the International Futures (IFs) system.

Almost all variables examined in Table 1 have relationship to TFP in the expected direction, even after controlling for logged GDP per capita, and significance values for the contributions are very often high even when the r-squared is not. To make the estimations consistent with the conditional convergence methodology, the coefficients for driver variables were computed in a two-step process: IVs were regressed against GDP per capita at PPP first and the residuals were used in subsequent regression with TFP. While recognizing the complications of identifying causality and its variation over time, we can average the coefficients and betas for each variable across the years with data shown in Table 1 to estimate the potential contribution to variation in TFP of each unit or standard deviation of the individual driver not "expected" at the country's level of GDP per capita.

**Table 1. Explanatory power for TFP level of IVs after control for GDP per capita at PPP.**

| Variable | Units | Year | N | Coefficient | T value | R squared | Beta | p value |
|---|---|---|---|---|---|---|---|---|
| Education Quality | Test score | 1990 | 28 | 0.10 | 2.09 | 0.14 | 0.33 | 0.09 |
| Education Quality | Test score | 2010 | 31 | 0.12 | 3.02 | 0.24 | 0.49 | 0.00 |
| Education Quality | Test score | 2015 | 111 | 0.06 | 3.09 | 0.08 | 0.28 | 0.00 |
| Average adult (15+) educational attainment | Years | 1990 | 114 | 0.12 | 2.44 | 0.05 | 0.23 | 0.00 |
| Average adult (15+) educational attainment | Years | 2000 | 113 | 0.08 | 2.03 | 0.03 | 0.17 | 0.00 |
| Average adult (15+) educational attainment | Years | 2010 | 113 | 0.13 | 3.86 | 0.07 | 0.26 | 0.00 |
| Average adult (15+) educational attainment | Years | 2015 | 111 | 0.13 | 2.20 | 0.04 | 0.20 | 0.00 |
| Average life expectancy at birth | Years | 1990 | 128 | 0.01 | 0.79 | 0.01 | 0.07 | 0.00 |
| Average life expectancy at birth | Years | 2000 | 128 | 0.01 | 1.49 | 0.00 | 0.54 | 0.00 |
| Average life expectancy at birth | Years | 2010 | 128 | 0.02 | 2.20 | 0.01 | 0.10 | 0.00 |
| Average life expectancy at birth | Years | 2015 | 111 | 0.02 | 1.34 | 0.02 | 0.12 | 0.00 |
| Education spending as a % of GDP | Percent | 2005 | 79 | 0.25 | 2.52 | 0.08 | 0.28 | 0.02 |
| Education spending as a % of GDP | Percent | 2010 | 85 | 0.25 | 2.68 | 0.08 | 0.28 | 0.02 |
| Education spending as a % of GDP | Percent | 2015 | 111 | 0.11 | 1.32 | 0.02 | 0.12 | 0.67 |
| Stunting | Percent | 2015 | 111 | -0.04 | -0.24 | 0.00 | -0.02 | 0.00 |
| Corruption (Transparency Index, Scale: 0–10) | Index | 2000 | 81 | 0.11 | 3.43 | 0.26 | 0.51 | 0.00 |
| Corruption (Transparency Index, Scale: 0–10) | Index | 2010 | 125 | 0.18 | 4.75 | 0.08 | 0.27 | 0.00 |
| Corruption (Transparency Index, Scale: 0–10) | Index | 2015 | 111 | 0.21 | 3.31 | 0.09 | 0.30 | 0.00 |
| Economic Freedom Index (Scale 1 to 10) | Index | 1990 | 91 | 0.30 | 2.50 | 0.06 | 0.26 | 0.00 |
| Economic Freedom Index (Scale 1 to 10) | Index | 2000 | 100 | 0.25 | -1.53 | 0.03 | 0.16 | 0.00 |
| Economic Freedom Index (Scale 1 to 10) | Index | 2010 | 120 | 0.19 | 1.11 | 0.01 | 0.10 | 0.00 |
| Economic Freedom Index (Scale 1 to 10) | Index | 2015 | 111 | 0.24 | 1.49 | 0.02 | 0.14 | 0.00 |
| Government Effectiveness (Scale 0 to 5) | Index | 2000 | 127 | 0.53 | 12.39 | 0.10 | 0.32 | 0.00 |
| Government Effectiveness (Scale 0 to 5) | Index | 2010 | 127 | 0.34 | 4.44 | 0.05 | 0.23 | 0.00 |
| Government Effectiveness (Scale 0 to 5) | Index | 2015 | 111 | 0.45 | 2.99 | 0.07 | 0.27 | 0.00 |
| Polity Index (Scale -10 to 10) | Index | 1990 | 85 | 0.01 | 0.56 | 0.00 | 0.06 | 0.00 |
| Polity Index (Scale -10 to 10) | Index | 2000 | 100 | 0.06 | 2.91 | 0.08 | 0.28 | 0.00 |
| Polity Index (Scale -10 to 10) | Index | 2010 | 103 | 0.06 | 2.68 | 0.06 | 0.26 | 0.00 |
| Polity Index (Scale -10 to 10) | Index | 2015 | 111 | 0.09 | 3.63 | 0.10 | 0.33 | 0.00 |
| R&D spending as a % of GDP | Percent | 2015 | 111 | 2.70 | 2.39 | 0.04 | 0.22 | 0.00 |
| Tertiary Education (Science/Eng Share) | Percent | 2000 | 49 | -0.045 | -1.11 | 0.03 | -0.16 | 0.01 |
| Tertiary Education (Science/Eng Share) | Percent | 2015 | 111 | -0.03 | -1.28 | 0.01 | -0.12 | 0.01 |
| Traditional Infrastructure Index (Scale -2 to 2) | Index | 1990 | 80 | 1.26 | 1.63 | 0.03 | 0.18 | 0.00 |
| Traditional Infrastructure Index (Scale -2 to 2) | Index | 2000 | 101 | 1.18 | 3.16 | 0.09 | 0.30 | 0.00 |
| Traditional Infrastructure Index (Scale -2 to 2) | Index | 2010 | 57 | 1.17 | 3.01 | 0.14 | 0.38 | 0.00 |
| Traditional Infrastructure Index (Scale -2 to 2) | Index | 2015 | 111 | 0.21 | 1.02 | 0.01 | 0.01 | 0.00 |
| Other infrastructure Spending as a % of GDP | Percent | 2005 | 15 | -0.22 | -0.90 | 0.06 | -0.24 | 0.17 |
| Other infrastructure Spending as a % of GDP | Percent | 2015 | 111 | -0.02 | -0.21 | 0.00 | -0.02 | 0.00 |

Note: Logged GDP per capita was entered as first variable in forward regression; statistics for it are not shown. Some IV values are not available for earlier years. "Other infrastructure" refers to spending not in categories explicitly represented in IFs and linked to other IVs (paved roads, electricity, water, sanitation, and information/communication technology).

Source: IFs Version 7.49 based on many data sources.

**Addressing relationships among the IVs.** The above discussion of Table 1 does not fully resolve issues around simultaneous parameterization of the contributions of multiple drivers to TFP. Control for GDP per capita does not remove all correlation among the IVs. One

potential aide to addressing remaining correlation is grouping the IVs. A principal components analysis or PCA (elaborated in S4 Appendix) reinforced the potential. It identified two groupings that correspond to two long used in the productivity formulation of IFs and identified as human capital (such as health and education) and sociopolitical capital (especially governance quality). It gives considerably more limited support, however, to two others used in this project and shown in Table 2. The first of those is physical capital, including the supportive infrastructure of society and the price of energy (sudden rises, as in the 1970s, can render some physical capital uneconomical, as suggested in Fig 2). The second is knowledge capital, including R&D spending, tertiary education, and openness to trade, variables with a prominent place in research on endogenous productivity growth.

The four categories together cut across the major theoretical and empirical traditions for understanding productivity. Much of the empirical work in the neoclassical tradition, tied to the Solow model, has focused on human capital. Much of the effort building on endogenous growth theory has emphasized what we here call social capital, in combination with the capacity for transfer of and ultimately development of knowledge and technology [48]. The addition in IFs of attention to country-level rather than just sectoral physical capital corrects a common omission in both sets of studies, with three references to studies of infrastructure in Durlauf, *et. al.* [30: 655] showing strong relationship with growth. Thus, the PCA, while offering support for two very important categories of productivity drivers, proved less useful than desired when looking more broadly at variables identified in the theoretical and empirical studies of productivity in the literature.

Even the four-category schema omits potential variables representing geography and culture that still other analyses have identified as important in productivity and economic growth, at times suggesting that they compete in a "horse race" against institutional variables [48]. Geography is a relative constant, however, so it may help explain levels of productivity but contributes little in most forecasting beyond representation in fixed-effect terms. Culture changes slowly but does change over the very long horizons of interest to us, and factors such as air conditioning and irrigation can even alter the impact of geography, so it may be useful to later add these to the set of IVs.

Exploring further for multicollinearity, Table 2 shows relationships of the IV residuals within and across the four categories. In general, it verifies that the risk of overestimation in formulations with multiple drivers is greater within than across the categories. For example, the highest r-squared of driver residuals across categories in Table 2 is 0.32 between life expectancy and government effectiveness. Yet, within the social capital category the r-squared of the relationship between the residuals of government transparency (the reverse of corruption) and government effectiveness reaches 0.74. Use of the categories therefore does have utility in grouping related variables and is maintained in IFs.

There are alternative ways of dealing with the remaining multicollinearity. One would be to omit some of the highly intercorrelated variables (which is done, as discussed below). Another would be to develop an index within each category to address the closer relationship of IVs within than across the categories; the PCA proved less useful in doing that than hoped. Again, in policy analysis we want to undertake interventions in the model around what-if questions focusing on the individual impact of a great many drivers—even government transparency and effectiveness, very highly correlated, *can and do* vary independently, arguing for an approach that attempts to maintain both in the structure.

**Table 2. Correlations in 2010 of the residual relationships among driving variables after control for GDP per capita (r-squared values).**

| | | Human Capital | | | | | Social Capital | | | | Knowledge Capital | | Physical Capital |
|---|---|---|---|---|---|---|---|---|---|---|---|---|---|
| | | Education Quality | Education Years | Life Expectancy | Education Expenditure | Stunting | Corruption | Economic Freedom | Government Effectiveness | Polity Democracy | Trade | R&D | Infrastructure Index |
| **Human Capital** | Education Quality | - | | | | | | | | | | | |
| | Education Years | 0.30 | - | | | | | | | | | | |
| | Life Expectancy | 0.05 | 0.38 | - | | | | | | | | | |
| | Education Expenditure | 0.67 | 0.00 | 0.02 | - | | | | | | | | |
| | Stunting | 0.00 | 0.33 | 0.19 | 0.00 | - | | | | | | | |
| **Social Capital** | Corruption | 0.20 | 0.15 | 0.16 | 0.00 | 0.04 | - | | | | | | |
| | Economic Freedom | 0.20 | 0.13 | 0.19 | 0.00 | 0.02 | 0.31 | - | | | | | |
| | Government Effectiveness | 0.30 | 0.32 | 0.32 | 0.00 | 0.10 | 0.74 | 0.42 | - | | | | |
| | Polity Democracy | 0.09 | 0.17 | 0.11 | 0.00 | 0.04 | 0.20 | 0.14 | 0.23 | - | | | |
| **Knowledge Capital** | Trade | 0.00 | 0.00 | 0.00 | 0.00 | 0.00 | 0.01 | 0.01 | 0.00 | 0.00 | - | | |
| | R&D | 0.30 | 0.21 | 0.07 | 0.00 | 0.03 | 0.18 | 0.07 | 0.31 | 0.02 | 0.01 | - | |
| **Physical Capital** | Infrastructure Index | 0.30 | 0.27 | 0.25 | 0.00 | 0.07 | 0.17 | 0.05 | 0.24 | 0.03 | 0.00 | 0.21 | - |
| | Infrastructure Spending | 0.04 | 0.00 | 0.01 | 0.94 | 0.00 | 0.00 | 0.00 | 0.00 | 0.00 | 0.00 | 0.00 | 0.01 |

Note: Trade openness values here also control for GDP per capita even though IFs introduces them relative to initial conditions and moving average, not GDP per capita. Infrastructure index is the sum of traditional (road transport, electricity, water, and sanitation) and ICT indices. Infrastructure spending is other than traditional or ICT. Table does not include other components of IFs approach not using GDP per capita control: disability, vocational education contribution (human capital), conflict (social capital), energy price (physical capital), and tertiary education (knowledge capital).

Source: IFs Version 7.49.

## Aggregating insights and building a model structure for TFP

In building the model formulation, other research and stylized facts drawn from the empirical and theoretical literatures augmented this project's own analysis (see Durlauf, *et al.* [49] on the benefits of model averaging methods and Durlauf [50] on the use of stylized facts). But the information from Tables 1 and 2 provides a foundation for specifying appropriate parameters in the relationship linking the IV residuals to TFP change.

An important challenge not much addressed in discussion to this point is relationship specification of conditional convergence impacts in the context of a dynamic, recursive structure with annual time steps of models like IFs. To illustrate, Table 1 showed that a 1-unit improvement in the residual value (controlling for the log of GDP per capita) of the 11-point corruption scale was associated with a 0.17 unit increase on the TFP scale. Yet, we would not expect such a *ceteris paribus* reduction in corruption to increase TFP that much in the year following decrease and possibly not for quite a few years. Although comparative statics analysis needs not address lags, policy analysis does, one reason for using the dynamic recursive IFs system.

Unfortunately, the literature on drivers of development is seldom of much help on temporal dynamics. As Durlauf, et al. [30: 630] emphasized:

> A common failing of panel data studies based on within-country variation is that researchers do not pay enough attention to the dynamics of adjustment. There are many panel data papers on human capital and growth that test only whether a change in school enrollment or years of schooling has an immediate effect on aggregate productivity, which seems an implausible hypothesis.

Those authors pointed to studies that do before and after analyses of major events as one way to better understand both magnitude of impact and dynamics. The statistical focus on TFP level as the dependent variable rather than on immediate or arbitrarily-lagged changes in TFP both forces the issue of attention to temporal dynamics and provides a bridge to it.

Table 3 builds on Table 1, links its information to the productivity parameterization in IFs, and facilitates understanding of temporal impacts. Columns A through D build on the statistical analysis of the project by summarizing some of it already reported (specifically, the contents in Column B based on Table 1) and by adding additional information from supporting analysis to other columns. Columns E through H turn to the actual parameters specified in IFs (Column E), drawing upon insights from the first set of columns (especially Column B from which the values seldom differ greatly) and on years of IFs-project attention more qualitatively to extensive literatures associated with most drivers and to the behavior of the model with interventions. The remaining columns in the right half of the table explore the temporal implications of the parameters.

More specifically, within the first set, Column (A) shows standard deviations (SDs) of the residuals of IVs after control for GDP per capita. The SDs facilitate analysis of relative impact of drivers with very different scales. Column B contains the average coefficients that cross-sectional analysis in multiple years found between the residuals of drivers and TFP. Column C scales the coefficients to indicate the percentage change in TFP associated with a single unit of each IV; it divides the residuals by the average value (3.2) of TFP (the Solow residual) globally in 2015 (the model base year) and converts the result to percentage terms. Column D indicates the percentage impact on TFP that each standard deviation of the residuals has, thereby facilitating comparison across drivers of unit-free contributions to TFP of changes in them.

Columns E through H move the attention to the parameters in IFs, their immediate impact on TFP in each time step, and their longer-term impact if residuals relative to the GDP per

**Table 3. Parameterization and temporal dynamics in the IFs productivity representation.**

| Independent Variable (IV) | Values Based on Comparative Cross-Sectional Analysis | | | | Values Using IFs Parameterization and Cross-Sectional Analysis to Consider Temporal Impact | | | |
|---|---|---|---|---|---|---|---|---|
| | A) Standard deviation (SD) in 2010 of independent variable residuals after control for GDP per capita | B) Coefficient of impact on TFP per unit of residual IV (average across years shown in Table 1) | C) Percentage impact on TFP per unit of residual IV at average TFP of 3.2 (B/3.2) | D) Percentage impact on TFP per SD of residual IV at average TFP of 3.2 (A*C) | E) Parameter in IFs (annual percent change in TFP per residual unit) | F) Annual percent change in TFP for 1 SD of residual IV in IFs (A*E) | G) Years in IFs to reach estimated impact at average TFP of 1 SD residual IV (D/F) | H) Years in IFs needed to advance TFP by 10% if maintain 1 SD residual (10/(F*100)) |
| Education Quality | 7.13 | 0.09 | 2.8% | 20.1% | 0.10% | 0.71% | 28.1 | 14 |
| Education Years | 2.82 | 0.12 | 3.8% | 10.6% | 0.20% | 0.56% | 18.8 | 18 |
| Life Expectancy | 8.68 | 0.02 | 0.6% | 5.4% | 0.00% | 0.00% | NA | NA |
| Education Spending % of GDP | 1.63 | 0.2 | 6.3% | 10.2% | 0.00% | 0.00% | NA | NA |
| Stunting | 7.89 | -0.04 | -0.1% | -1.0% | -0.04% | -0.32% | 3.1 | 32 |
| Corruption (11-point transparency scale) | 2.10 | 0.17 | 5.3% | 11.2% | 0.20% | 0.42% | 26.6 | 24 |
| Economic Freedom Index (1 to 10) | 0.73 | 0.25 | 7.8% | 5.7% | 0.25% | 0.18% | 31.3 | 55 |
| Government Effectiveness (6-point scale) | 0.99 | 0.44 | 13.8% | 13.6% | 0.50% | 0.50% | 27.5 | 20 |
| Polity Index (Scale -10 to 10) | 5.92 | 0.06 | 1.9% | 11.1% | 0.00% | 0.00% | NA | NA |
| R&D Spending as % of GDP | 0.12 | 1.72 | 53.8% | 6.5% | 1.50% | 0.18% | 35.8 | 56 |
| Tertiary Education (science/engineering share) | 21.79 | -0.04 | -1.3% | -27.2% | 0.01% | 0.22% | NA | 46 |
| Traditional Infrastructure Index (-2 to 2) | 0.45 | 0.96 | 30.0% | 13.5% | 1.00% | 0.45% | 30.0 | 22 |
| Other Infrastructure Spending % of GDP | 1.90 | -0.12 | -3.8% | -7.1% | 0.15% | 0.29% | NA | 35 |

Notes: See Table 1 and related discussion of regression analysis of IV residuals (controlling for GDP per capita) against TFP. IVs with parameters of 0.0 in Column E have typically been omitted from IFs analysis for reasons discussed earlier but made available for scenario use. The year 2015 was used for SDs of other infrastructure spending and for R&D spending because of scarcity of data in earlier years. Prior to the analysis of this table parameters in IFs differed somewhat from above: education quality was 0.15; stunting was -0.025; economic freedom was 0.1; research and development was 0.5; traditional infrastructure was 2.0; other infrastructure spending was 0.1. *The analytical results of this table have enhanced IFs by supplementing earlier parameter values rooted almost entirely in qualitative literature analysis.*
Source: IFs Version 7.49.

capita-based, structurally-expected values persist. Specifically, Column E shows the Base Case scenario parameter in IFs that converts a single unit of the IV residual to *annual percentage point change* in TFP during model runs. Although heavily influenced by the statistically estimated values in Column B, differences represent analysis of literature and model behavior plus

other qualitatively-based adjustments related to multicollinearity and basic reasonableness explained below. Column F uses the standard deviations of Column A and the IFs parameters to compute the annual percentage change that a full SD of residual variation would contribute to TFP given the parameters in IFs. This helps clarify the relative impacts of the IVs in IFs and again can be very useful in the project's own comparison of those impacts with the highly diverse but still useful literature on contributions to growth.

Column G further ties the IFs parameterization to the project's statistical analysis by computing the number of years that each SD of IV residuals would need to persist in IFs to move TFP by the percentage calculated in Column D as the full impact of each SD. Thus, that column shows comparative IV impacts and the temporal playing out of TFP dynamics with IFs parameterization. Stunting is an outlier in the table on the low side only because the low coefficient from the statistical analysis generated a low percentage impact of a full SD of the IV; the statistical analysis on stunting, based on only a single year of relatively sparse data, is not strong. Column H builds the temporal analysis further by telling us how many years in IFs it would take for each standard deviation of IV to raise TFP by an arbitrary 10%; the comparison across drivers facilitates still another basic check on reasonableness.

While parameterization within IFs has looked to the literature extensively [24] as well as to this new research, judgment remains important. Sometimes sensitivity analysis of the model in comparison with insights from real-world cases encouraged small adjustments, a variation of event analysis.

Multicollinearity across drivers remains an issue. In another step to address it, the IFs project has assigned default parameters of 0.0 for some obviously spurious or redundant IVs. For instance, the clear redundancy of life expectancy and education spending with other human capital IVs (see Table 2 again) and the literature's mixed findings with respect to democracy's contribution to growth led to Base Case assignment of 0.0 on each.

Two additional features, related to the dynamic behavior of the resultant structure in IFs, merit comment. First, a push on any driving variable via scenario analysis that moves it above expected values at a country's level of GDP per capita will contribute to the indicated increase in TFP growth. Because that increase will also raise GDP per capita, the expected value of the driving variable will rise in future years, cutting back future impetus from it unless dynamics underlying change in the driving variable continue its positioning above expected values. Also, a rise in GDP per capita driven by one IV will raise expected values for and reduce residual impact of other driving variables if their level does not change, another control on multicollinear impacts. Further, the argument in the structural development literature that variables shaping development tend to advance simultaneously suggests that a disproportionate residual level of any one IV relative to others (e.g. a policy push greatly raising education but ignoring corruption reduction) is likely subject to diminishing returns. Algorithmic specification in IFs slightly dampens the contribution of individual IV residuals that considerably exceed others.

Second, the system appropriately produces *percentage change* in TFP, which has a direct relationship to percentage change in economic growth. The *absolute levels of annual change* in TFP will, of course, vary greatly across countries and time. Consider, for instance, Iceland and Burundi. The TFP value (the Solow residual) in the 2015 model base year is 8.50 for the former and 0.76 for the latter. An increase by one in average years of adult education relative to expected values in each country will result in the same percentage increase in TFP and very different absolute increases. Convergence of TFP values in Burundi to those in Iceland depends on a combination of the inverted-U shaped function of basic convergence described earlier (giving Burundi some relative boost even when education years are those expected at their development level), and it depends on any additional increase in years of education or other IVs relative to expected values that Burundi might achieve.

As this four-step discussion of methodology has emphasized, structure and parameterization of the multivariate formulation for representing TFP advance in a dynamic IAM is challenging. Any representation must be open to change, subject to improved data, to enhanced statistical analysis, and to sensitivity and scenario analysis.

## Results

The obvious analytical question is: how well does the integrated, multivariate TFP system perform in a base or reference run of the model and in other scenario analysis? Subsections briefly summarize performance of the system in three arenas:

1. Comparing long-term projections from the IFs system with the OECD quantification of TFP in the SSPs. While focusing here only on Base Case projections from IFs in comparison with the five SSPs from the OECD and especially SSP2 (the "Middle of the Road" scenario), the IFs system would allow exogenous introduction of other SSP variable quantifications including population, education, and inequality, as well as scenario elaboration of other qualitative scenario elements such as governance. Thus, the current analysis could subsequently be extended to compare TFP from IFs replications of other SSPs and to literature on climate change impacts [51].

2. Gaining insight into differential economic prospects of countries in the short-term.

3. Exploring the implications of different sets of policy-related interventions for integrated pursuit of the SDGs, taking advantage of having represented both the drivers of TFP and their growth impacts. Other research without the methodological elaboration of this study has used prior IFs versions for analysis of selected SDG subsets [52, 53] and in issue-area specific comparison of projections with those from other IAMs [24]. Analysis here looks broadly across the SDGs in a more mid-term analysis.

### Comparison of IFs projections with SSP projections from the OECD

Dellink, *et al.* [20] produced TFP and GDP projections at the OECD for the Shared Socioeconomic Pathways (SSPs). Dellink and Château shared their TFP projections for all five SSPs. As noted earlier, that project also used a conditional convergence structure and production functions with Cobb-Douglas structure. The OECD approach to representing TFP convergence toward a shifting frontier involves distance from the frontier, trade openness, and country-specific fixed effects. Its production function additionally represents extraction and processing of oil and gas and autonomous energy efficiency. It integrates education advance with labor as human capital. (See also Château, *et al.* [54] on the OECD ENV-Linkages model as the foundation for its work).

How different is the Base Case scenario of IFs with its more elaborate representation of TFP drivers from the SSP projections of the OECD, particularly from SSP2? SSP2 is the scenario closest to the IFs Base Case in terms of storyline and is often labeled Middle of the Road. Potentially, a TFP structure driven by a very wide range of IVs, each in turn endogenously computed in other models of IFs system that also respond to economic growth in an integrated, hard-linked system with two-way causality, could produce very different behavior.

Fig 3 shows global projections of TFP from the OECD across all five SSPs and the Base Case projection from IFs (using the Base Case version before IFs began to represent the impacts of COVID-19 to enhance comparability with the OECD projections). The initialization of TFP in IFs projections using this project's calculation of the Solow residual means that it includes in TFP the human capital development variables that the OECD approach puts with the labor

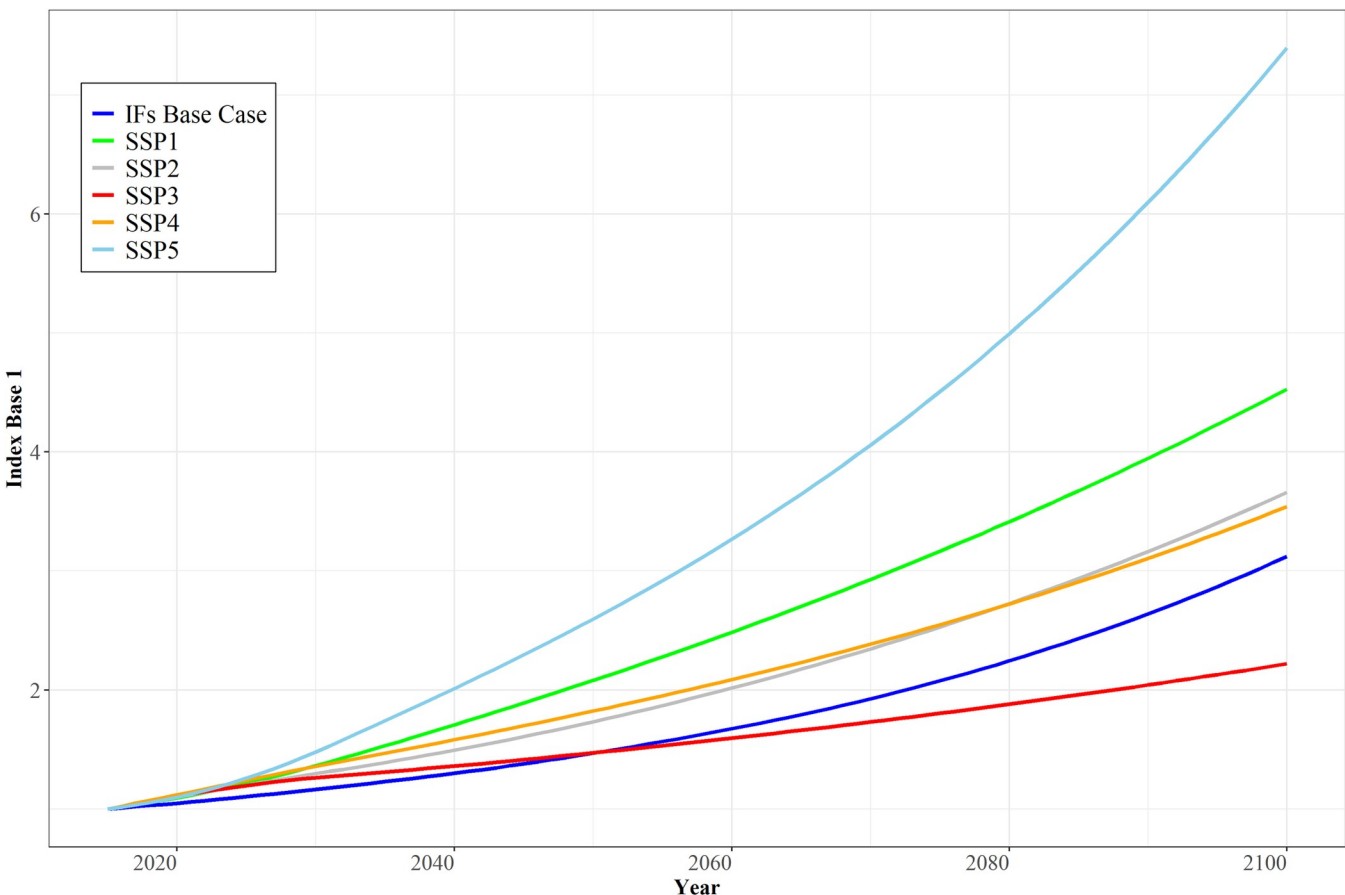

**Fig 3. Global TFP projections from the OECD across SSPs and the Base Case projection from IFs.** Notes: TFP scaling in the two sources differs and has been indexed to 1 to facilitate comparison. Country aggregation uses simple averages rather than GDP-weighting. SSPs 2 and 4 and the IFs Base Case are so close that the lines nearly overlap. Source: OECD projections courtesy of Rob Dellink and Jean Château; IFs Version 7.61.

term. Scaling of values for TFP from both projects to 1. 0 in 2015 compensates for that and other initial condition differences, facilitating comparison. The TFP growth projection from IFs runs parallels to but somewhat below the OECD SSP2 scenario (and similarly close to SSP4, which is nearly identical and often labeled Inequality).

Fig 4 turns attention from global TFP patterns to the relative advance in higher- and lower-income country sets and thus to the issue of global inequality, using OECD and non-OECD countries as proxies for rich and poor. Here the IFs Base Case scenario differs more from the OECD SSP2 projection. The exogenous use by IFs of IMF GDP data and estimates through 2021 somewhat reduces the TFP convergence pattern of non-OECD countries in those early years relative to SSP scenarios. Subsequently, the IFs Base Case ratio of productivity in the two country sets tends to trace a pattern with acceleration of catch-up in the first half of the century, influenced heavily by the impacts on TFP in Africa, Latin America, and developing Asia of narrowing global gaps in human capital, including health and education, and of advances in social capital including governance capacity and quality.

In the second half of the century that catching up continues but narrowing of the gap with systemic productivity leaders inevitably slows it considerably. The resultant global inequality pattern in general is between SSP2 and SSP4 (Inequality). PWT historical data show that after a long, slowly increasing divergence between the TFP levels of OECD and non-OECD

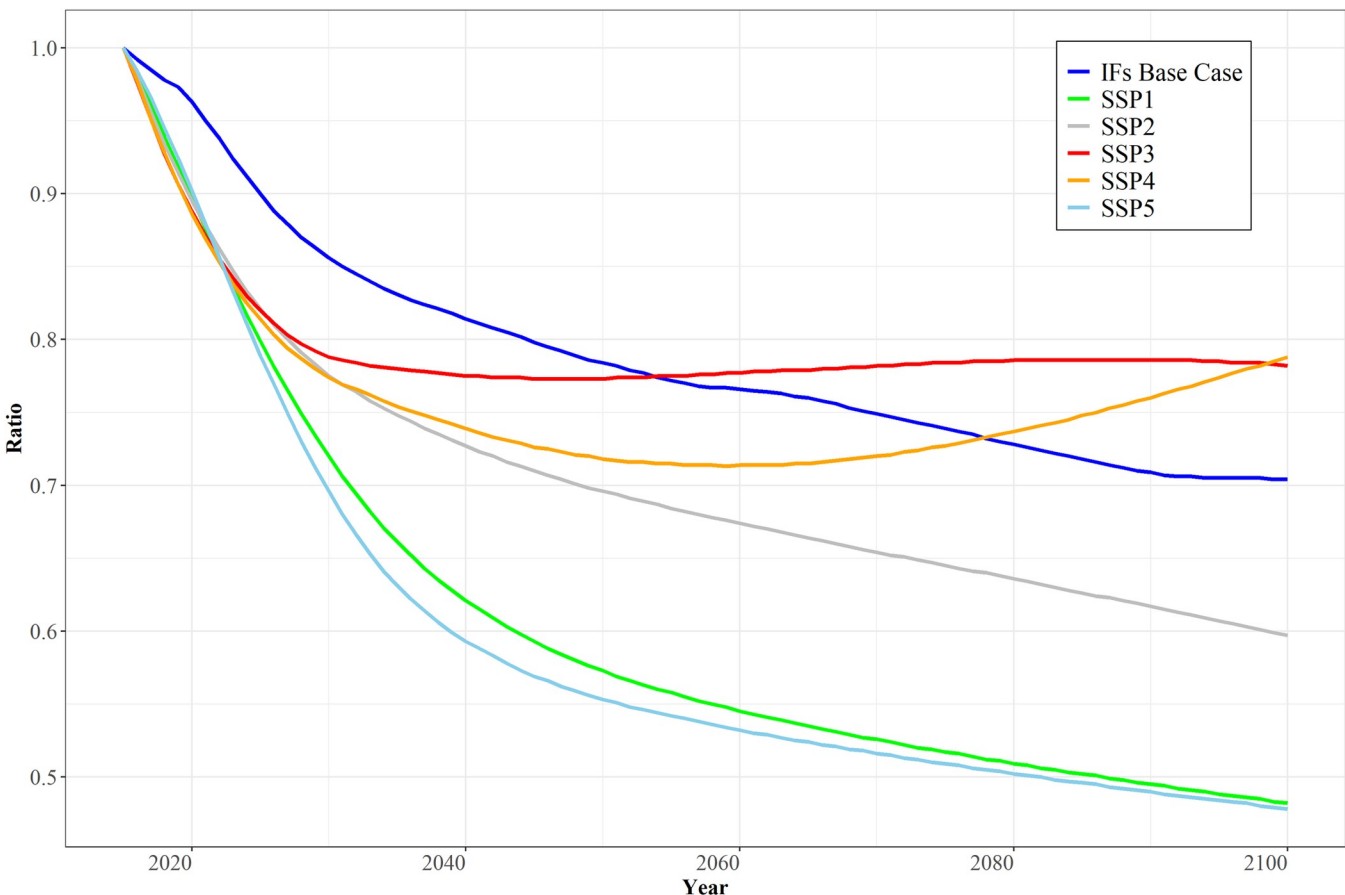

**Fig 4. Ratios of OECD and non-OECD TFP projections in the OECD SSP scenarios and in the Base Case projections from IFs.** Notes: Country aggregation uses GDP-weighting. Ratios are indexed to 1 in 2015 facilitate comparison across the models; absolute OECD values remain significantly above those in non-OECD countries. Source: OECD courtesy of Rob Dellink and Jean Château; IFs Version 7.61.

countries until about 2000, the ratio of values in the two country sets fell by roughly one-third between 2000 and 2014. Thus, a similar decline in that ratio through 2100 in the IFs Base Case would seem a conservative projection (SSP2 generates about another 10 percent narrowing).

## Gaining insight into prospects for differential economic growth

One value added promised by extensive endogenization of TFP and its drivers should be in assessment of how well countries are positioned also in the considerably shorter run for strong economic growth that can reduce poverty and bring other advantages. Fig 5 shows actual annual GDP growth rates from 2010 through 2020 for two groupings of 10 countries, namely those that IFs calculates from data in 2015 to have the highest and lowest potential rates of productivity increase. The calculations that determine the two groups take into account the four sets of drivers identified in this article, the conditional convergence pattern across GDP per capita levels, and 2015 calculation of the Solow residual increase from the Cobb-Douglas production function. The rates of GDP growth in the figure for those two country sets are values from the IMF's *World Economic Outlook* prior to the pandemic. The growth rate differential shown in the IMF data and estimates is striking, and it might well be larger than many observers would expect *a priori* from the membership of the two country sets. The hypothesized high-TFP growth set consists of Azerbaijan, Cambodia, China, Georgia, Ireland, Lithuania, the Maldives,

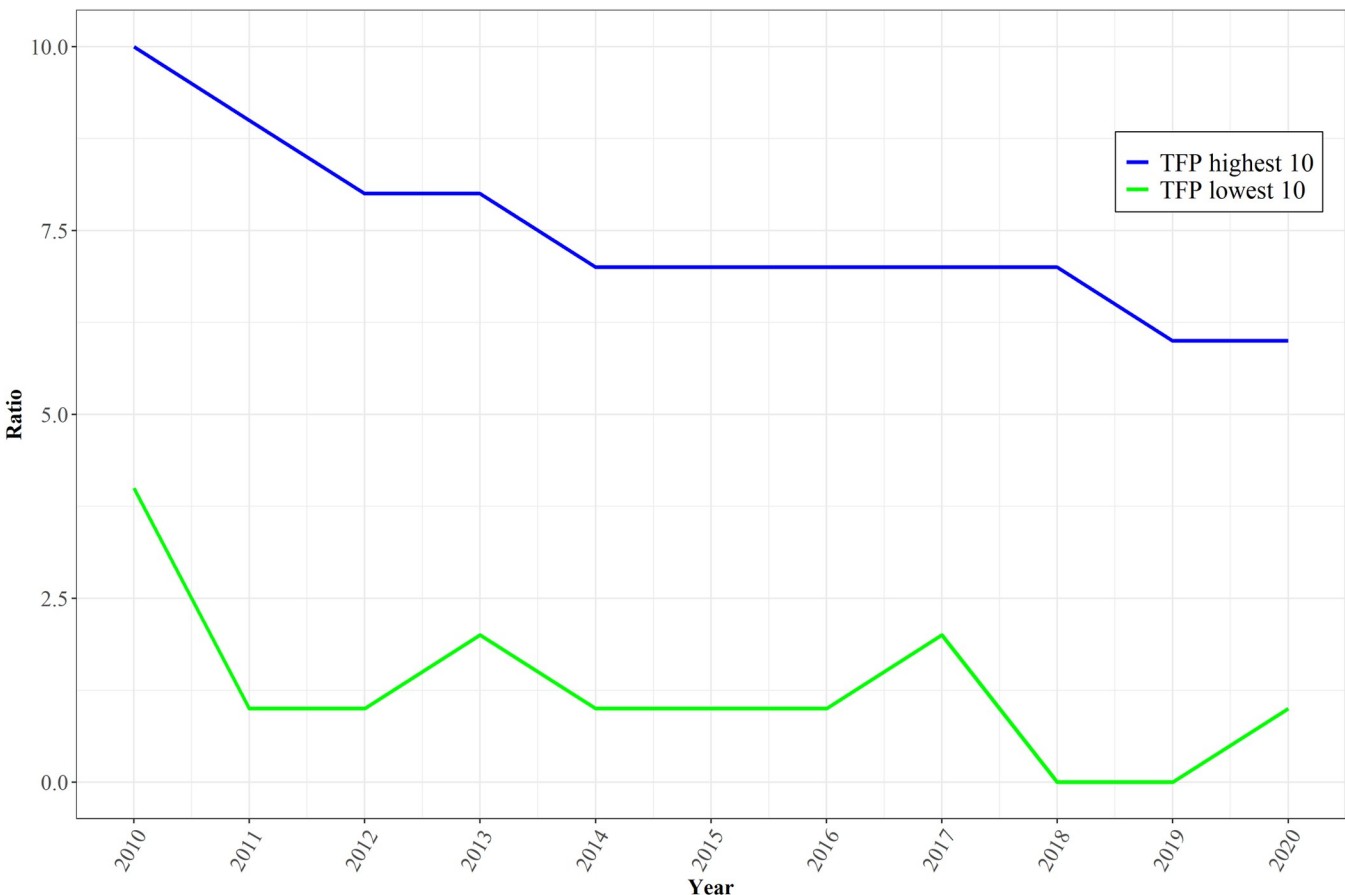

**Fig 5. Actual and IMF-estimated GDP growth rates of countries with highest and lowest productivity growth calculations by IFs in 2015.** Note: 2019 and 2020 were pre-pandemic estimates, other years are data. Source: IFs Version 7.61 and IMF World Economic Outlook 2019.

Mongolia, Myanmar, and Turkmenistan. The low-TFP growth rate set consists of Algeria, the Republic of the Congo, France, Haiti, Iran, Japan, Lesotho, Mauritius, Suriname, and Zambia. In projections of the IFs Base Case and other scenarios the country membership of such sets will vary across time depending on the multiple drivers in the productivity formulations.

## SDG pursuit considering synergies and trade-offs

The examples above suggest the insight into longer- and shorter-term country productivity and growth prospects offered by an integrated analysis system with extensively endogenized TFP. Such potential can be extended to mid-range analysis of policy interventions in pursuit of a large set of aims like the SDGs. Some of the numerous trade-offs and synergies inherent in such analysis involve financial constraints around revenues and expenditures, and their treatment benefits not only from endogenous TFP representation but from the IFs system's endogenization of a social accounting matrix so that there are "no free lunches" in directing resources to specific goals. SDG analysis with the IFs system also benefits from its inclusion of a broad range of socioeconomic and biophysical models (see S4 Appendix), extending backward and forward linkages of variables well beyond the general equilibrium economic model and its production function. Nonetheless, the production function is a key point at which many variables interact and push forward economic dynamics that in turn affect other SDG-related variables.

Ongoing work at the Pardee Center for International Futures uses IF to explore the contribution to goal achievement from different individual and packaged interventions. Illustratively, one analysis [55] looks at the individual and collective implications of two clusters of quite aggressive interventions, but ones scaled in examination of best practice across countries historically. Those clusters are a Human Development (HD) focused scenario, a Natural System Sustainability (NSS) emphasis scenario, as well as their linking in a Combined SDG (CSDG) scenario, all in comparison with the Current Path (CP) scenario (another label for the Base Case scenario).

Fig 6 shows the current global status of progress toward one key target on each of the goals and the projected status in 2050 in each of the four scenarios—2030 is too near to illustrate much intervention potential, and few goals will likely be universally attained even by 2050, especially those focused on biophysical sustainability. The analysis does find that synergies dominate relative to trade-offs in integrated analysis. A not surprising exception is control of carbon emissions, where advance on human development in the absence of interventions directed also at sustainability can limit or even reverse progress.

## Discussion

This article has not minimized the difficulties in building an integrated, extensively endogenized representation of economic productivity and integrating it into the framework of a larger model system or suite. The TFP approach builds upon a core or basic long-term productivity convergence pattern related to GDP per capita at purchasing power parity (PPP), which is understood to be conditional. The conditionality is a function of a wide range of independent variables that all tend to progress together in structural relationship with the economic development process. It is thus the impact of the residuals of those other IVs in their own relationships with GDP per capita (higher or lower than "expected" values) that determine faster or slower advance than that associated with GDP per capita alone. The overall structure and parameterization of the TFP representation in IFs looks to empirical work from the literature

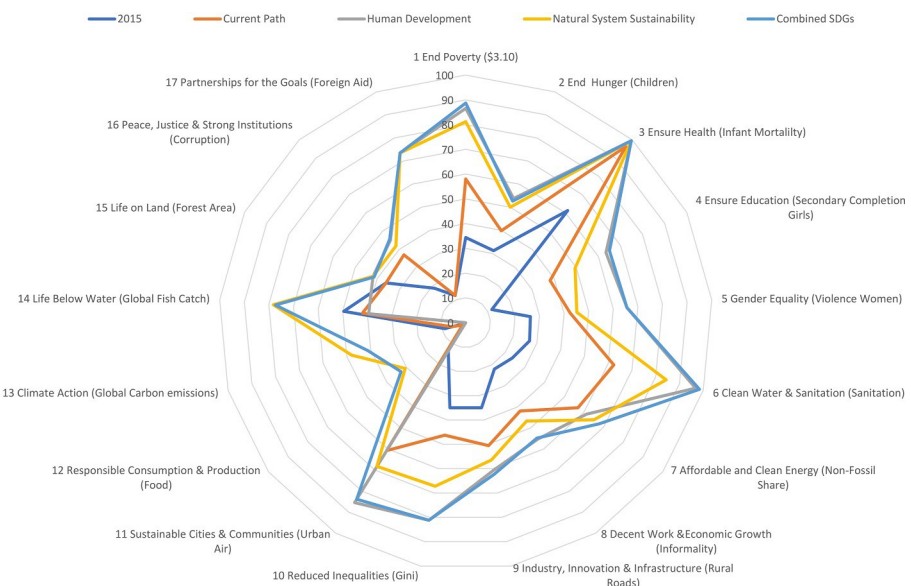

**Fig 6. Analysis of progress toward SDG targets on current path and with three sets of interventions.** Note: Values are the percentage of countries reaching the identified target in 2015 or 2050 scenarios, except fisheries and carbon where they are extent of progress toward a global goal. Source: Hughes [55]; IFs Version 7.45.

and temporal dynamics of IFs behavior as well as to the statistical analysis of the project. Judgment has been required to deal with the complications introduced to data analysis by the needs (1) to build and use a historical data series of TFP with great volatility and cyclical behavior over time; (2) to address the significant multicollinearity within a very large set of independent variables within and across clusters even after control for GDP per capita; and (3) to deal with the historical variability and thus increased uncertainty in temporal dynamics of independent variable impact. Much more might be done around these issues.

The article has also suggested the utility of the functioning TFP system within IFs via comparison with other long-term projections of TFP globally, in a glance at insight with respect to shorter-term prospects of countries, and with a look at how such a system can help in extensive, integrated analysis across the SDGs. The ability of the TFP system described to draw upon inputs from many issue-area models and in turn to feed forward growth implications to those same models is an important step needed in highly integrated analysis around sustainable development.

Refinement and extension of the work needs be undertaken. The intention of this article is to provide quantitative support to the parameterized approach for calculating TFP in IFs. As additional data and the evolving literature provide the basis, the statistical methodology and parameters related to TFP can be refined. S6 Appendix provides information about access to IFs and replication of the analysis. The openness of the system, including the ability to change all parameters of the production function and the broader internal capabilities in IFs for supporting data and scenario analysis, can support refinement and extension. Policy analysis obviously benefits from the kind of integration described in this report, so the motivation for continued improvement is great.

## Supporting information

**S1 Appendix. Understanding and measuring TFP.**
(DOCX)

**S2 Appendix. Data sources for TFP driver analysis.**
(DOCX)

**S3 Appendix. Temporally-sliced cross-sectional analysis.**
(DOCX)

**S4 Appendix. The International Futures (IFs) model system.**
(DOCX)

**S5 Appendix. Principal components analysis.**
(DOCX)

**S6 Appendix. Software and data availability for use and replication.**
(DOCX)

**S1 File.**
(DOCX)

**S2 File.**
(RMD)

**S3 File.**
(RMD)

**S4 File.**
(IPYNB)

**S1 Data.**
(XLSX)

**S2 Data.**
(CSV)

**S3 Data.**
(CSV)

## Acknowledgments

We much appreciate the provision of OECD productivity projections for the SSPs by Rob Dellink and Jean Château, feedback on the manuscript by Jonathan Moyer, Brian O'Neill, and Mickey Rafa, and the openness of the International Futures (IFs) integrated assessment model system for development and use. The IFs system itself is the result of many important contributions, not least those by José Solórzano, Mohammod Irfan, David Bohl, and Steve Hedden.

## Author Contributions

**Conceptualization:** Barry B. Hughes.

**Data curation:** Barry B. Hughes, Kanishka Narayan.

**Formal analysis:** Barry B. Hughes, Kanishka Narayan.

**Investigation:** Barry B. Hughes, Kanishka Narayan.

**Methodology:** Barry B. Hughes, Kanishka Narayan.

**Resources:** Barry B. Hughes, Kanishka Narayan.

**Software:** Barry B. Hughes, Kanishka Narayan.

**Validation:** Barry B. Hughes, Kanishka Narayan.

**Visualization:** Barry B. Hughes, Kanishka Narayan.

**Writing – original draft:** Barry B. Hughes.

**Writing – review & editing:** Barry B. Hughes, Kanishka Narayan.

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
