## [Decision Letter · Decision Letter 0]

1 Dec 2020

PONE-D-20-15458

Enhancing integrated analysis of national and global goal pursuit by endogenizing economic productivity

PLOS ONE

Dear Dr. Hughes,

Thank you for submitting your manuscript to PLOS ONE. After careful consideration, we feel that it has merit but does not fully meet PLOS ONE’s publication criteria as it currently stands. Therefore, we invite you to submit a revised version of the manuscript that addresses the points raised during the review process.

We look forward to receiving your revised manuscript.

Kind regards,

Yangyang Xu

Academic Editor

PLOS ONE

Journal Requirements:

Reviewers' comments:

Reviewer's Responses to Questions

**Comments to the Author**

1. Is the manuscript technically sound, and do the data support the conclusions?

Reviewer #1: Yes

Reviewer #2: Yes

Reviewer #3: Yes

2. Has the statistical analysis been performed appropriately and rigorously? 

Reviewer #1: Yes

Reviewer #2: Yes

Reviewer #3: Yes

3. Have the authors made all data underlying the findings in their manuscript fully available?

Reviewer #1: Yes

Reviewer #2: Yes

Reviewer #3: Yes

4. Is the manuscript presented in an intelligible fashion and written in standard English?

Reviewer #1: Yes

Reviewer #2: Yes

Reviewer #3: Yes

5. Review Comments to the Author

Reviewer #1: The paper offers significant insights to the process of endogenizing total factor productivity (TFP) in global modeling, including its relevance for making sound projections on future developments with regards to Sustainable Development Goals (SDGs) development on the national scale. The research thus complements existing research on modeling integrated pathways, e.g. as pursued under the framework of the shared socioeconomic pathways (SSPs). The research is also novel as it presents new model formulations and the rationale behind using the suggested formulas in comparison with other modeling formulations. It further connects the importance of extending beyond TFP measures and GDP in social-ecological systems modeling, and how it relates to the pursuit of reaching the 2030 Agenda’s Sustainable Development Goals.

A question for the author is how the suggested formulation relate to similar efforts made by the Millennium Institute’s formulation of the Threshold 21 iSDG system dynamics-based model, in which TFP is endogenized for the respective economic sectors (agriculture, industry and services). This is documented by Pedercini: https://www.millennium-institute.org/documentation, in relation to SSPs discussed in Allen et al. 2019 and discussed in relation to SDG synergies in Pedercini et al. 2020 (references below). The statistical sophistication and argumentation behind the calculations of TFP in International Futures and what it entails perhaps differentiates it from the Millennium Institute’s iSDG model, but this is something that needs some elaboration by the authors. Alternatively, if the authors do not think the Millennium Institute-related work on similar matters is relevant, it would be interesting to see some reflections on this.

Overall, I assess it as an excellent research article that well deserves publication in PlosOne.

Minor comments: The Figure 4 is not entirely clear, caption could be made clearer. The paper is sometimes referred to as a ”report”, maybe this is on purpose. For me as a non-native speaker it seems like “paper” or “article” would be a better word.

References

Allen, C., G. Metternicht, T. Wiedmann, and M. Pedercini. 2019. Greater gains for Australia by tackling all SDGs but the last steps will be the most challenging. Nature Sustainability 2(11):1041–1050.

Pedercini, M., G. Zuellich, K. Dianati, and S. Arquitt. 2018. Toward achieving Sustainable Development Goals in Ivory Coast: Simulating pathways to sustainable development. Sustainable Development 26:588–595.

Pedercini, M., S. Arquitt, D. Collste, and H. Herren. 2020. Harvesting synergy from sustainable development goal interactions. Proceedings of the National Academy of Sciences 116(46):23021–23028.

Reviewer #2: Review for “Enhancing integrated analysis of national and global goal pursuit by endogenizing economic productivity”.

General comments:

This manuscript presents modifications to the International Futures (IF) Integrated Assessment Model that endogenize economic productivity expressed as total factor productivity (TFP). The contribution of this work is very high and important to society insofar as understanding how economic growth might be affected by various policies, in the context of large global changes, will help us navigate challenging policy dilemmas. Overall, I recommend this for publication assuming the minor revisions, discussed in more depth below, are made (excepting any comment listed as not required).

The point made on line 65 of computing TFP and economic growth as a function AND a driver of related variables is important but not explained with enough detail. To me, this was not an important part of the narrative and I would suggest (though not require) removing.

It would be worth clarifying when the unmodified IF model is discussed vs. the IF-with-changes model to make clear that the TFP endogenization has been included in the IF model used to populate table 3.

Some sections seemed unnecessarily long, e.g. paragraphs at 246 and 254.

Need more explanation on line 291 for; “Ifs facilitates such scenario analysis with flexible parameterization of temperature change impact on TFP as advanced by structurally-related factors.”

The PCA near line 335 was VERY interesting and perhaps can be more showcased.

I wonder, though definitely do not require, how the use of recent machine learning techniques, such as elastic net regression, could augment the section discussing multicollinearity and to do automated variable selection in place of manually creating parsimony.

Table 3 is a central part of the paper but it is not explained clearly. Explain the split grouping of columns a-d and e-h. Careful reading of prior sections does technically explain it, but this was time consuming and could be made more clear. Consider also synthetic comments that summarize the table coefficients for policy makers: e.g., “Column F in this table implies that a policy that improved variable X by Y would have a Z impact on TFP.

Minor Comments

Line 74: Very nicely selected set of literature links.

Line 89: “considerably further extends.” Consider rewording.

Lin2 212: “the core representation is ceteris paribus,” Consider adding a comma.

Reviewer #3: This is an interesting and important paper. Here are a few comments:

- The discussion of why GDP per capita would be a poor instrumental variable would benefit from a explicit discussion of the clear failures of the exclusion restriction

- It seems that non-classical measurement error could be an issue here, particularly if measurement error is correlated within countries across indicators

- The authors typically control linearly for log GDP per capita in their models. But it is not obvious that a linear control is the correct specification. It would be helpful to either defend this assumption or show robustness to alternate functional form specifications, such as quadratics or ideally binned versions. Even then, given the correlation between log GDP per capita and the independent variables of interest in, for instance, table 1, the most appropriate statistical approach would be something like Oster (2019) using the change in the magnitude of the coefficient on the IV after having added log GDP per capita and adjusting for the change in the explanatory power of the model (r2). Otherwise, the actual coefficients on the IV vs. log GDP per capita may just be due to differences in noise between the two variables. Instead, the authors residualize the IV on log GDP per capita, which seems to go against the intuition of the paper which is that there is meaningful variation in the IVs that stems directly from income. By residualizing, this feedback is essentially turned off, which seems odd given the main thesis.

- Given that the PCA analysis identifies essentially two groups of predictors, within which the IVs are very correlated, the authors' critique of the existing literature seems a bit strange. It seems that just using two predictors could therefore perform quite well?

- It would be interesting to think about how spillovers across countries impact these results. There are spillovers through technology, financial links, and general the global economy. Thus errors in the regression will not be i.i.d., but especially for this specification, the residualized IVs will change.

- The paper would benefit from more tests of whether the model is successful or not. How can the reader know whether the model fits well? How can we be confident in the predictions for the future? More explicitly laying out the assumptions would also be useful.

6. PLOS authors have the option to publish the peer review history of their article (what does this mean?). If published, this will include your full peer review and any attached files.

Reviewer #1: **Yes: **David Collste

Reviewer #2: **Yes: **Justin A Johnson

Reviewer #3: No

---

## [Author Response · Author response to Decision Letter 0]

1 Jan 2021

Reviewer #1: The paper offers significant insights to the process of endogenizing total factor productivity (TFP) in global modeling, including its relevance for making sound projections on future developments with regards to Sustainable Development Goals (SDGs) development on the national scale. The research thus complements existing research on modeling integrated pathways, e.g. as pursued under the framework of the shared socioeconomic pathways (SSPs). The research is also novel as it presents new model formulations and the rationale behind using the suggested formulas in comparison with other modeling formulations. It further connects the importance of extending beyond TFP measures and GDP in social-ecological systems modeling, and how it relates to the pursuit of reaching the 2030 Agenda’s Sustainable Development Goals.

A question for the author is how the suggested formulation relate to similar efforts made by the Millennium Institute’s formulation of the Threshold 21 iSDG system dynamics-based model, in which TFP is endogenized for the respective economic sectors (agriculture, industry and services). This is documented by Pedercini: https://www.millennium-institute.org/documentation, in relation to SSPs discussed in Allen et al. 2019 and discussed in relation to SDG synergies in Pedercini et al. 2020 (references below). The statistical sophistication and argumentation behind the calculations of TFP in International Futures and what it entails perhaps differentiates it from the Millennium Institute’s iSDG model, but this is something that needs some elaboration by the authors. Alternatively, if the authors do not think the Millennium Institute-related work on similar matters is relevant, it would be interesting to see some reflections on this.

The question raised is a very good one. This lead author is very much aware of and appreciative of the iSDG system and Threshold 21 before it (having known Jerry Barney since the 1970s and Global 2000 and being very saddened by his death this year). In another manuscript currently under review, focused on integrated analysis of the SDGs as opposed to the treatment of TFP, I point to iSDG as a rare tool that can help us analyze them in in a fully integrated, systemic manner. Our omission of reference here was inappropriate and has been remedied at the end of the Introduction (including a reference to an iSDG paper by Collste et al. already identified in that other paper, now also under review). A difficulty for us in this paper is that the multi-driver treatment of productivity in iSDG is described very generally in its documentation as involving normalization assuming Hick-neutral technological change. The normalization reference sounds as if it most likely involves a scaling of a weighted product of the drivers, but we have found no elaboration of rules for driver combination. Hence, I think the methods in our work are quite different (in fact, more connected to quantitative and qualitative analysis and approaches that are standard in economic modeling), but we lack basis for elaborating any comparison.

Overall, I assess it as an excellent research article that well deserves publication in PlosOne.

Minor comments: The Figure 4 is not entirely clear, caption could be made clearer. The paper is sometimes referred to as a ”report”, maybe this is on purpose. For me as a non-native speaker it seems like “paper” or “article” would be a better word.

Yes, the title on Figure 4 was confusing and has been edited. I have always hesitated to refer to a manuscript as an “article” during the review stage but have now presumptively changed “report” to “article” throughout the revised paper. Thanks for the specific references; we chose the 2nd Pedercini et al article as the one having more information on the model but pointed also to the on-line documentation of iSDG.

References

Allen, C., G. Metternicht, T. Wiedmann, and M. Pedercini. 2019. Greater gains for Australia by tackling all SDGs but the last steps will be the most challenging. Nature Sustainability 2(11):1041–1050.

Pedercini, M., G. Zuellich, K. Dianati, and S. Arquitt. 2018. Toward achieving Sustainable Development Goals in Ivory Coast: Simulating pathways to sustainable development. Sustainable Development 26:588–595.

Pedercini, M., S. Arquitt, D. Collste, and H. Herren. 2020. Harvesting synergy from sustainable development goal interactions. Proceedings of the National Academy of Sciences 116(46):23021–23028.

Reviewer #2: Review for “Enhancing integrated analysis of national and global goal pursuit by endogenizing economic productivity”.

General comments:

This manuscript presents modifications to the International Futures (IF) Integrated Assessment Model that endogenize economic productivity expressed as total factor productivity (TFP). The contribution of this work is very high and important to society insofar as understanding how economic growth might be affected by various policies, in the context of large global changes, will help us navigate challenging policy dilemmas. Overall, I recommend this for publication assuming the minor revisions, discussed in more depth below, are made (excepting any comment listed as not required).

The point made on line 65 of computing TFP and economic growth as a function AND a driver of related variables is important but not explained with enough detail. To me, this was not an important part of the narrative and I would suggest (though not require) removing

Agreed and done. It was clumsy.

It would be worth clarifying when the unmodified IF model is discussed vs. the IF-with-changes model to make clear that the TFP endogenization has been included in the IF model used to populate table 3.

 Added to the notes for Table 3.

Some sections seemed unnecessarily long, e.g. paragraphs at 246 and 254.

Agreed on removal of the first paragraph; some of the text from the second has been repurposed. 

Need more explanation on line 291 for; “Ifs facilitates such scenario analysis with flexible parameterization of temperature change impact on TFP as advanced by structurally-related factors.”

Again, that was clumsy and inadequate; reworked.

The PCA near line 335 was VERY interesting and perhaps can be more showcased.

We struggled to figure out how to use PCA more fully because it seemed like a very useful way in which to structure relationships within and across the dimensions of interest to us. But as the paragraphs that follow the one to which you point indicate, there was only a partial match between the dimensions that emerged from our efforts with it and the theoretically/conceptually useful ones in literature and long-standing analyses. Hence, we referred to it in the text because of the useful partial match and its potential utility in analysis like ours, but we ultimately relegated our more extended efforts to the annex. The text concerning the PCA has been strengthened.

I wonder, though definitely do not require, how the use of recent machine learning techniques, such as elastic net regression, could augment the section discussing multicollinearity and to do automated variable selection in place of manually creating parsimony.

Whoa. A fascinating idea. But as you implicitly suggest, probably an entirely different project and paper. Something for us to keep in mind (and learn how to do).

Table 3 is a central part of the paper but it is not explained clearly. Explain the split grouping of columns a-d and e-h. Careful reading of prior sections does technically explain it, but this was time consuming and could be made more clear. Consider also synthetic comments that summarize the table coefficients for policy makers: e.g., “Column F in this table implies that a policy that improved variable X by Y would have a Z impact on TFP.

We have significantly edited the text explaining the table, including the suggested addition of an introductory explanation of the two column groupings. The combination in that table of statistical basis, actual IFs parameterization tied to that but also influenced by literature and behavioral analysis, and some insight into the implications of the parameterization for temporal dynamics does complicate its understanding. Hopefully the editing will help walk the reader through what we agree is a central part of the paper. Thank you for urging that be done. 

Minor Comments

Thank you for the compliment on Line 74; the other two suggestions were addressed in edits.

Line 74: Very nicely selected set of literature links.

Line 89: “considerably further extends.” Consider rewording.

Line 212: “the core representation is ceteris paribus,” Consider adding a comma.

Reviewer #3: This is an interesting and important paper. Here are a few comments:

Author general comments in response: It seems useful to address the important comments of this reviewer mostly as a set (we give the final two some additional comment), rather than individually as was done for those from the first two reviewers. In doing so, we can first acknowledge and appreciate that the reviewer obviously has great statistical sophistication and has raised important issues concerning the work of the project.

The comments and questions pushed us to extend and enhance substantially our textual explanation of what we are doing and how we are attempting to do it. Our hope is that the enhancements are on target with respect to most of the issues raised by the reviewer. There are three interrelated points on which we focused textual changes related to the first several issues raised by the reviewer. Please see especially the substantial editing in Section 2.3 named “Understanding multiple interacting contributions to TFP change” and starting at line 227, within its introduction and across its subsections:

1. Making clear that we recognize the great difficulties (at least for us with our own capabilities and we think more generally) of undertaking and relying on statistical techniques in building an integrated representation of the forces in the development process that drive TFP advance when that representation involves a very large number of highly interrelated developmental variables. Already in the introduction to that section we explicitly recognize some of the challenges to and limitations in our own approach. Our belief, as expressed there and elaborated subsequently, is that our work is still useful in itself as well as foundational for more advanced work.

2. Emphasizing the great many variables individually and collectively that have importance to policy analysts. The difficulty was compounded for us when efforts such as those with principal components analysis failed even to draw out fully the dimensions across those variables that theoretical and empirical literatures have identified as important. Our statistical analysis is intended to help build a representation of TFP dynamics that also needs to look to those literatures and to the dynamic behavior of the model. We believe that building an extensively integrated representation of TFP, drawing on a great many policy-relevant variables, will always require statistical analysis (and that ours can be improved upon), but also qualitative judgment often tied to existing literature and model behavior. We have tried to make that combination clearer in the text and to recognize the limitations of our own approach. 

3. Arguing that our basic model structuring and quantification approach has important merit that we have tried to explain more fully. As development economists have long recognized, addressing development structurally benefits from a foundational representation of economic growth and/or productivity advance as conditionally driven by GDP per capita. Other driving variables structurally advance with that GDP per capita (in pretty clearly logged relationships with GDP per capita in our analysis); their contribution to the conditionality lies only in their advance at rates faster or slower than that which normally characterizes their relationship with GDP per capita. Hence our use of the residuals from their relationship with GDP per capita in representing those contributions. In the forecasting of the model, various dynamics and policy interventions will, of course, change the annual computation of those residuals and therefore their impact over time. Again, we acknowledge clearly that future work can improve many elements of the methodology.

Overall, the reviewer comments made clear to us the need to explain better our approach and the reasons for it. Quite extensive textual changes have attempted to do that. In the process we have also backed out some weak and seemingly unnecessary discussion, such as that of instrumental variables given that we don’t use them. Our general comments here and the manuscript changes have, we hope, usefully addressed the comments and questions below. Please look to the last two for brief additional comment.

- The discussion of why GDP per capita would be a poor instrumental variable would benefit from a explicit discussion of the clear failures of the exclusion restriction

- It seems that non-classical measurement error could be an issue here, particularly if measurement error is correlated within countries across indicators

- The authors typically control linearly for log GDP per capita in their models. But it is not obvious that a linear control is the correct specification. It would be helpful to either defend this assumption or show robustness to alternate functional form specifications, such as quadratics or ideally binned versions. Even then, given the correlation between log GDP per capita and the independent variables of interest in, for instance, table 1, the most appropriate statistical approach would be something like Oster (2019) using the change in the magnitude of the coefficient on the IV after having added log GDP per capita and adjusting for the change in the explanatory power of the model (r2). Otherwise, the actual coefficients on the IV vs. log GDP per capita may just be due to differences in noise between the two variables. Instead, the authors residualize the IV on log GDP per capita, which seems to go against the intuition of the paper which is that there is meaningful variation in the IVs that stems directly from income. By residualizing, this feedback is essentially turned off, which seems odd given the main thesis.

- Given that the PCA analysis identifies essentially two groups of predictors, within which the IVs are very correlated, the authors' critique of the existing literature seems a bit strange. It seems that just using two predictors could therefore perform quite well?

- It would be interesting to think about how spillovers across countries impact these results. There are spillovers through technology, financial links, and general the global economy. Thus errors in the regression will not be i.i.d., but especially for this specification, the residualized IVs will change.

Yes, it is very true that there are intercountry spillover effects that affect TFP directly and via impacts on driving variables. Section 2.2 (named “Representing technological leadership and basic convergence” and starting on line 186) explains the representation of technological leadership and basic or core convergence in IFs of other countries converges toward TFP of the leader, a high-level explicit representation of DV intercountry spillover (implicitly responsive to IV drivers). More generally, however, we admit to not having given appropriate thought to spillovers specific to the IVs driving conditional convergence (even the current global emphasis on universal primary and then secondary education is effectively one such; the very existence of the SDGs is one even more broadly ideational). The descriptions of IFs in text and Appendix S4 help identify some of the more direct and material intercountry linkages such as trade, FDI, foreign aid, and migration. Trade openness is one of the independent variables discussed and represented in IFs. Although now referring very generally to such spillovers in new text, we have not picked up on their implications for statistical analysis.

- The paper would benefit from more tests of whether the model is successful or not. How can the reader know whether the model fits well? How can we be confident in the predictions for the future? More explicitly laying out the assumptions would also be useful.

To help establish model validity, we undertook three different types of analysis/tests, including comparing the projections of TFP in IFs with similar projections from the OECD (all in Section 3 “Results” starting on line 534). More can, of course, always be done on the validity support issue. In fact, over much time we have done many historical runs of IFs from 1960 and other base years forward, quite satisfactorily comparing behavior with data. In other work we are picking up that thread and will be reporting results elsewhere, but we chose not to dig into that very large in-progress project in this paper.

---

## [Decision Letter · Decision Letter 1]

27 Jan 2021

Enhancing integrated analysis of national and global goal pursuit by endogenizing economic productivity

PONE-D-20-15458R1

Dear Dr. Hughes,

We’re pleased to inform you that your manuscript has been judged scientifically suitable for publication and will be formally accepted for publication once it meets all outstanding technical requirements.

Kind regards,

Yangyang Xu

Academic Editor

PLOS ONE

Additional Editor Comments (optional):

Reviewers' comments:

Reviewer's Responses to Questions

**Comments to the Author**

1. If the authors have adequately addressed your comments raised in a previous round of review and you feel that this manuscript is now acceptable for publication, you may indicate that here to bypass the “Comments to the Author” section, enter your conflict of interest statement in the “Confidential to Editor” section, and submit your "Accept" recommendation.

Reviewer #1: All comments have been addressed

Reviewer #2: All comments have been addressed

Reviewer #3: All comments have been addressed

2. Is the manuscript technically sound, and do the data support the conclusions?

Reviewer #1: (No Response)

Reviewer #2: Yes

Reviewer #3: Yes

3. Has the statistical analysis been performed appropriately and rigorously? 

Reviewer #1: (No Response)

Reviewer #2: Yes

Reviewer #3: Yes

4. Have the authors made all data underlying the findings in their manuscript fully available?

Reviewer #1: (No Response)

Reviewer #2: Yes

Reviewer #3: Yes

5. Is the manuscript presented in an intelligible fashion and written in standard English?

Reviewer #1: (No Response)

Reviewer #2: Yes

Reviewer #3: Yes

6. Review Comments to the Author

Reviewer #1: (No Response)

Reviewer #2: All comments of mine have been nicely addressed. This is ready for publication. In particular, the edits to the main table now allow for clearer interpretation.

Reviewer #3: I am grateful to the authors for their thoughtful response and consideration of my comments and congratulate them on a great paper!

7. PLOS authors have the option to publish the peer review history of their article (what does this mean?). If published, this will include your full peer review and any attached files.

Reviewer #1: **Yes: **David Collste, Stockholm Resilience Centre, Stockholm University

Reviewer #2: **Yes: **Justin A Johnson

Reviewer #3: No

---

## [Editor Report · Acceptance letter]

3 Feb 2021

PONE-D-20-15458R1 

Enhancing integrated analysis of national and global goal pursuit by endogenizing economic productivity 

Dear Dr. Hughes:

I'm pleased to inform you that your manuscript has been deemed suitable for publication in PLOS ONE. Congratulations! Your manuscript is now with our production department. 

Kind regards, 

on behalf of

Dr. Yangyang Xu 

Academic Editor

PLOS ONE